# Comprehensive Evaluation of the Nutritional Properties of Different Germplasms of *Polygonatum cyrtonema* Hua

**DOI:** 10.3390/foods13060815

**Published:** 2024-03-07

**Authors:** Mei Lu, Luping Zhang, Shixin Kang, Fengxiao Ren, Luyun Yang, Qingyou Zhang, Qiaojun Jia

**Affiliations:** 1College of Life Sciences and Medicine, Zhejiang Sci-Tech University, Hangzhou 310018, China; 13862044607@163.com (M.L.); z2359224830@163.com (L.Z.); ksx13832336990@163.com (S.K.); renfengxiao2022@163.com (F.R.); 18957252806@163.com (L.Y.); qiyo0226@163.com (Q.Z.); 2Key Laboratory of Plant Secondary Metabolism and Regulation of Zhejiang Province, Zhejiang Sci-Tech University, Hangzhou 310018, China

**Keywords:** *Polygonatum cyrtonema*, nutrient content, nutritional quality, germplasm

## Abstract

*Polygonatum cyrtonema* Hua, an edible resource and medical material, is mainly consumed as a food in China. However, few published studies have comprehensively assessed its nutritional components. In this study, the proximate, carbohydrate, and dietary fiber contents as well as the mineral, vitamin, and amino acid compositions of five sources of *P. cyrtomena* grown in Yuhang district, Hangzhou city, Zhejiang province, were investigated. The nutritional profile of the five germplasms was investigated using analytical chemistry methods. All germplasms had a low starch content and contained greater amounts of carbohydrates (23.25–34.29%), protein (2.96–5.40%), Ca (195.08–282.08 mg/100 g), Fe (29.68–59.37 mg/100 g), and vitamin C (60.49–149.86 mg/100 g) in comparison to ginger, yam, and potatoes. The polysaccharide content ranged from 16.92% to 28.48%, representing the main source of carbohydrates. Fructose, a desirable sweetener, was the most abundant monosaccharide, representing 1.06% to 4.88% of the content. *P. cyrtonema* was found to be high in dietary fiber, with pectin and resistant starch being the major soluble components and hemicellulose being the dominant insoluble dietary fiber. A correlation analysis (CA) revealed significant correlations for the carbohydrate components and dietary fiber fractions with other nutrients. A principal component analysis (PCA) identified significant differences between the nutritional characteristics of the five germplasms, with Huanggang having the highest comprehensive quality scores. Moreover, ten nutrient components were selected as potential indicators that could be used to further evaluate the nutritional quality of *P. cyrtomena*. Our results demonstrate the rich nutrient composition and characteristics of *P. cyrtonema* and provide a valuable reference for the future development and utilization of *Polygonatum*.

## 1. Introduction

*Polygonatum* species are perennial herbs from the Asparagaceae family that are widely distributed in China and have been used as medicine and food sources for more than 2000 years. *Polygonatum cyrtonema* Hua, a species from this genus that was introduced in the *Chinese Pharmacopoeia* (2020 edition) [1], is mainly distributed in the middle and lower reaches of the Yangtze River, including the Guizhou, Zhejiang, Hunan, Hubei, and Sichuan provinces [2], where it is planted in the forest and does not occupy arable land [3]. *Polygonati rhizoma*, commonly known as *Huangjing* in China, is the dried rhizome from several *Polygonatum* species, including *P. cyrtonema* Hua (the State Pharmacopoeia Committee of China, 2020), and has been used as a substitute for food since ancient times [4]. *Mingyi Bielu* documented that the long-term consumption of *Huangjing* can strengthen the body, delay aging, and eradicate hunger [5]. *Baopuzi Neipian* recorded that *Huangjing* can replace grain to combat hunger during times of famine [6]. Modern research has revealed that *Huangjing* contains abundant active ingredients such as saponins, flavonoids, polysaccharides, and lectins [7,8]. However, polysaccharides are the only component specified in the pharmacopoeia [1]. *Huangjing* polysaccharides lower blood sugar and blood lipid contents, show antitumor and anti-inflammatory activities, relieve fatigue and aging, improve immunity, and regulate the intestinal microbial community [9,10,11,12]. With the aging of the population and the shift in dietary needs from eating well to eating nutritiously and healthily, *Polygonatum* has gained attention due to its numerous health advantages.

Currently, approximately 80% of *Huangjing* produced is consumed as food in China, with an annual demand ranging from 3500 to 4000 tons [13,14]. Rhizomes are commonly steamed and sundried nine times to make preserves, and there are multiple *Huangjing* health items on the market, including wine, tea, yogurt, biscuits, cream, and tablets [9]. Due to its active ingredients, health products to resist aging, decrease fatigue, enhance immunity, regulate blood sugar, and promote sleep have been developed [14].

The nutrient composition of *Huangjing* includes 2.23–39.54% polysaccharides [15,16,17], 3.82–11.81% protein [18,19], 0.13–4.03% fat [19,20], and 1.58–9.80% ash [19,21]. The carbohydrate (17.5–68.01%) and vitamin C (0.5–170 mg/100 g) contents of the rhizome vary significantly between species [18,21]. In addition, *Huangjing* is a rich source of K, Ca, Mg, Fe, and amino acids [22]. The available literature on the nutrients contained in *Huangjing* is limited and mainly focuses on polysaccharides. According to Si and Zhu (2021), polysaccharides, oligofructose, and fructose make up around half of the nutrients in the rhizome [3]. However, there are few reports concerning monosaccharides and oligofructose. A previous study showed significant variation in the polysaccharide content of 25 wild *P. cyrtonema* resources, ranging from 2.234% to 14.094% [16]. Similarly, a high variation (9.56–17.68%) in the polysaccharide content was observed in 18 wild *P. cyrtonema* located in the Hunan experimental forest [23]. In addition, dietary fiber is an important nutrient that promotes satiation and satiety [24], but its content and composition remain unclear.

While *Huangjing* has been reported to bring considerable benefits to farmers, with yields of 1500 kg of dried product per hectare after four years of cultivation [25] and a price after processing of USD 55.6/kg, roughly five times the cost of the raw materials (USD 10.43/kg) [3], people have begun to artificially plant *P. cyrtonema* to increase income. Jiao et al. [26] reported all new *Polygonatum* Mill. rhizomes produced by artificial planting had an increased polysaccharide content and met the *Chinese Pharmacopoeia* standard limits after. However, few studies have carried out a comprehensive nutritional quality evaluation of different germplasm artificially cultivated *Polygonatum* resources.

Thus, in order to explore the nutrient composition and provide a scientific basis for the further processing and utilization of *Polygonatum*, we carried out a comprehensive evaluation of the nutritional properties of different germplasms. Five *P. cyrtonema* germplasms introduced from Hubei, Hunan, and Anhui to Zhejiang were selected after three years of cultivation, and their nutrient compositions and nutritional quality were analyzed and evaluated. Then, a correlation analysis (CA) and principal component analysis (PCA) were employed to identify their nutritional differences and evaluate the nutritional quality of the *Polygonatum* germplasm.

## 2. Materials and Methods

### 2.1. Source of Materials and Preparation

Five wild *P. cyrtonema* germplasms were collected from different regions and cultivated in Yuhang district, Hangzhou city, Zhejiang province, China. Three were collected from Huanggang (Hubei province), Qingyang (Anhui province), and Chizhou (Anhui province), respectively and planted in the germplasm resource garden (common garden) in Baizhang town. The other two were collected from different districts in Yueyang, Hunan province and planted on two different family farms. The geographic and climatic characteristics of these locations are shown in Appendix A. Figure 1 shows a map of the provenance and planting location of the five *P. cyrtonema*. All samples were authenticated by Prof. Zongsuo Liang (Zhejiang Sci-Tech University), assigned voucher specimen numbers, and deposited at the Key Laboratory of Plant Secondary Metabolism and Regulation of Zhejiang Province. The voucher specimen number and specific information on the planting locations of the five *P. cyrtonema* types are listed in Table 1. All samples were harvested in November 2021 after three years of cultivation. The rhizomes were sliced, dried to a constant weight, and ground into a fine powder (100 mesh). The powder was packaged in sealed plastic bags and stored in dry glassware at room temperature until use.

### 2.2. Proximate and Carbohydrate Composition Analysis

The moisture content was determined using an oven-drying method (105 °C for 8 h), based on Chinese National Standards (CNS) GB 5009.3-2016 [27]. A macro-Kjeldahl method (N × 6.25) was used to evaluate the protein content, in accordance with CNS GB 5009.5-2016 [28]. The lipid content was measured using petroleum ether as the extractant in a Soxhlet apparatus (Hangzhou Mick Chemical Instrument Co., Ltd., Hangzhou, China) (CNS GB 5009.6-2016) [29]. The ash content was determined by weighing the samples before and after heat treatment in a muffle furnace (Jinan PRECISION&SCIENTIFIC Instrument Co., Ltd., Jinan, China) (550 °C for 5 h), in accordance with CNS GB 5009.4-2016 [30]. The polysaccharide content was determined by anthrone–sulfuric acid colorimetry as follows. First, 0.5 g of powder mixed with 10 mL of water was placed in a 75 °C water bath for 2.5 h; then, 40 mL of anhydrous ethanol was added to the supernatant and left overnight at 4 °C. Finally, the precipitate was dissolved in water for determination [31]. The carbohydrate content was obtained by combining the polysaccharide content with the glucose, fructose, and sucrose contents. The starch content was quantified using an acid–hydrolyzed starch assay kit (BC0705, Beijing Solarbio Science and Technology Co., Ltd., Beijing, China). Each sample was prepared in triplicate.

### 2.3. Fiber Composition Analysis

The resistant starch (RS) content was determined using the enzyme digestion method, as described in the AOAC methods [32]. The sample powder (0.1 g) and 4 mL of α-tryptic amylase suspension were added into a centrifuge tube, mixed thoroughly, and shaken at 37 °C for 16 h. Then, 4 mL of anhydrous ethanol and 8 mL of 50% ethanol were added to wash the precipitate, respectively. The precipitate was treated with 2 mL of 2 mol/L KOH, 8 mL of 1.2 mol/L sodium acetate buffer (pH 3.8), and 1 mL of amyl glucosidase (AMG) in a 50 °C water bath for 30 min, and the glucose content was determined using the 3,5-dinitrosalicylic acid (DNS) method. The resistant starch content of the sample was calculated by multiplying its glucose content by 0.9.

The pectin content was analyzed using the spectrophotometric method, in accordance with CNS NY/T 2016-2011 [33]. The powder (0.1 g) was placed in a centrifuge tube and mixed with anhydrous ethanol in an 85 °C water bath for 10 min. The precipitate was then washed continuously with anhydrous ethanol in a water bath at 85 °C until the Molisch reaction of sugar was no longer present in the supernatant. The precipitate was then mixed with 10 mL of distilled water and 0.5 mL of 40 g/L NaOH and agitated for 15 min before being filtered. Then, 1 mL of the filtrate, 0.25 mL of carbazole ethanol solution, and 5 mL of concentrated sulfuric acid were mixed and placed in an 85 °C water bath for 20 min. After quick cooling, the absorbance was measured at 525 nm. Galacturonic acid was used as the standard substance and quantified by the standard curve.

The cellulose content was determined by the method described by Zhao et al. (2021) [34]. The sample powder (0.05 g) was treated with a mixture of 5 mL of acetic acid and nitric acid and heated for 25 min; then, 10 mL of 10% sulfuric acid and 10 mL of 0.01 mol/L potassium dichromate were added, and the mixture was heated for 10 min. The precipitate from each treatment was washed with distilled water three times. The cellulose content was calculated by adding 5 mL of 20% potassium iodide and titrating with 1 mL of 0.5% starch as an indicator and 0.2 mol/L of sodium thiosulfate.

The hemicellulose content was determined, as described previously, as follows. First, 0.1 g of powder was boiled in 10 mL 80% calcium nitrate solution, the precipitate was rinsed three times with hot water, hydrolyzed with 2 mol/L of hydrochloric acid, and neutralized with sodium hydroxide solution, and the reducing sugar in the solution was determined with the DNS method [34].

The lignin content was determined using sulfuric acid hydrolysis methods as follows: First, 0.05 g of powder was treated with 1% acetic acid. Then, a mixture of 70% ethanol and 30% ether (1:1 by volume) was added, and the precipitate was dissolved in 72% sulfuric acid for 16 h and then mixed with distilled water and barium chloride solution. Then, the precipitate was washed twice in distilled water, and the lignin concentration was titrated with 0.2 mol/L of sodium thiosulphate by adding 5 mL of 20% potassium iodide and titrating with 1 mL of 1% starch as an indicator [34].

The national standard method CNS GB 5009.88-2014 was used to determine the total dietary fiber content [35]. After desugarizing, drying, and sieving, the duplicate specimens were digested with heat-stable α-amylase, protease, and glucoamylase, precipitated with 95% ethanol, and filtered. The residue was washed with 78% ethanol, 95% ethanol, and acetone and dried, and the residue was weighed. The protein and ash contents of the residue were determined, and the dietary fiber content was calculated using the formula.
$$Total\;dietary\;fiber\;(\%)=\frac{{\bar{m}}_{R}-m_{P}-m_{A}-m_{B}}{\bar{m}\times \frac{m_{C}}{m_{D}}}\times 100$$
where ${\bar{m}}_{R}$ = mean weight of double sample residues, $m_{P}$ = weight of protein in sample residue, $m_{A}$ = weight of ash in sample residue, $m_{B}$ = weight of empty, $\bar{m}$ = mean weight of double samples, $m_{C}$ = weight of sample before desugarizing, and $m_{D}$ = weight of sample after desugarizing.

### 2.4. Monosaccharide and Oligosaccharide Analysis

The monosaccharide and oligosaccharide concentrations were determined using high-performance liquid chromatography with evaporative light scattering detection (HPLC-ELSD), which were slightly modified from previous studies [36,37]. Each sample (0.05 g) was extracted using distilled water (1.8 mL) with ultrasonication (100 W) for 1 h. After that, the extracts were centrifuged (12,000 r/min for 10 min) to separate the supernatant from the residues. The extracts of small-molecule saccharides were passed through a 0.22 µm filter for HPLC-ELSD analysis. Standard solutions of D-glucose, D-fructose, sucrose, nystose, 1-kestose, and 1F-fructofuranosylnystose at appropriate concentrations were prepared for the construction of calibration curves by plotting the extracted chromatogram peak area versus the concentration.

Chromatographic analysis was carried out on a Waters e2695 liquid chromatography system coupled with a 2424 ELS detector (Waters Corporation, Milford, MA, USA). Acetonitrile (containing 0.02% triethylamine) and ultrapure water were used as mobile phases A and B, respectively. The solvent flow rate was 1.0 mL/min, the column (XBridge Amide 5 µm, 250 mm × 4.6 mm) was operated at 35 °C, and the injection volume was 15 µL. The solvent gradient was used as follows. For the first 10 min, the mobile phase composition transitioned from 85% A to 75% A; for 10–50 min, 75% A and 25% B were used as the mobile phases; and for 50–55 min, 75% A was gradually returned to 85% A. The ELSD conditions were as follows: the drift tube was kept at 50 °C, the drying gas (N_2_) had a flow rate of 40 psi, the nebulizer cooling mode was used, and the gain value was 1.

### 2.5. Mineral Analysis

The mineral content of each sample was determined by atomic absorption spectroscopy. Each sample (0.2 g) was mixed with nitric acid (10 mL) and hydrogen peroxide (2 mL) in a polytetrafluoroethylene sample cup. A digestion tank was assembled and placed in a microwave digestion instrument (Jinan Hanon Instrument Co., Ltd., Jinan, China) for digestion. Then, the digestion tanks were taken out for acid removal after cooling to 60 °C, and the digested sample was pipetted into a 50 mL volumetric flask and made up to the desired volume with distilled water. The sample solution was filtered through a 0.45 μm microporous membrane prior to analysis [38].

Manganese, zinc, iron, sodium, potassium, calcium, magnesium, copper, cadmium, lead, and aluminum were determined using an Agilent 240 flame atomic absorption spectrometer with the 120 Graphite Tube Atomizer (Agilent Technologies Inc., Santa Clara, CA, USA) [39]. A set of standard solutions for each element was prepared using the following concentrations: 0, 0.1, 0.2, 0.4, 0.8, 1.6, and 3.2 ppm or 0, 2, 4, 6, 8, 16, and 32 ppb. The standard solutions were tested, and their values were recorded. The appropriate cathode lamp was fixed for each element. The atomizer of the instrument was dipped into the sample solution, and a meter reading was taken. The values obtained from the standards were used to plot the calibration curve for each test element, and the concentrations of the sample element were determined through extrapolation from the graph as ppm or ppb off the curve.

### 2.6. Vitamin Determination

The vitamin C concentration was determined spectrophotometrically with fast blue B salt (FBSB), as described by Zhou and Lou (2004) [40]. First, 0.1 g of powder was mixed with 0.5 mL of 2 mol/L acetic acid, 0.2 mL of 0.25 mol/L LEDTA, and 4.3 mL of anhydrous ethanol and sonicated for 15 min (100 W). Then, 5 mL of water was added, and the mixture was sonicated for another 15 min. Next, 600 μL of the supernatant was mixed with 200 μL of water, 50 μL of 2 mol/L acetic acid, 100 μL of 0.25 mol/L LEDTA, and 50 μL of 0.6% FBSB. Finally, the absorbance was measured at 420 nm after centrifugation at 8500 r/min for 5 min. The carotenoid content was determined using the following method. First, 0.2 g of powder was mixed with 5 mL of ethyl acetate in a test tube, the mixture was placed in a 50 °C water bath for 50 min, and then the absorbance value of the supernatant was measured at 450 nm [41]. The thiamine content was obtained with the fluorescence spectrophotometric method described in CNS GB 5009.84-2016 [42]. The riboflavin content was detected using a vitamin B2 test kit (fluorescence spectrophotometry), which was purchased from Shanghai Yuanye Biotechnology Co., Ltd. (Shanghai, China).

### 2.7. Amino Acid Analysis

The free amino acid composition was examined by the Tea Quality Inspection and Supervision Centre, Ministry of Agriculture and Rural Affairs, P.R.C., using an automated amino acid analyzer (Sykam S433D, Sykam GmbH, Munich, Germany). First, fine powder (0.2 g) was extracted in a 90 °C water bath for 30 min with 20 mL of boiling water. The supernatant was collected and filtered through 0.22 μm hydrophilic nylon membrane filters, and the volume was made up to 20 mL with distilled water. Afterward, the solution was mixed 1:1 (*v*:*v*) with the sample diluent and stored at 4 °C for testing.

### 2.8. Statistical Analysis

Data represent the mean of three replication analyses, and the results are reported as the mean ± standard deviation. Statistical analysis was performed using IBM SPSS 25.0 Statistics (SPSS Inc., Chicago, IL, USA). A one-way analysis of variance (ANOVA) was used to determine whether there were any statistically significant differences (*p* < 0.05) between groups. A correlation analysis (CA) and principal components analysis (PCA) were performed using Origin 2021 (OriginLab, Northampton, MA, USA).

## 3. Results and Discussion

### 3.1. Proximate and Carbohydrate Compositions of P. cyrtonema

The proximate and carbohydrate compositions of the five different *P. cyrtonema* germplasms are shown in Table 2. The moisture content of the dry samples ranged from 4.28% for Yueyang2 to 5.39% for Huanggang, which is consistent with the results reported by Sun et al. (2022) [19]. The protein content varied among the five germplasms, and the average value (4.52%) was similar to previously reported results for *P. cyrtonema* from Guizhou and Sichuan provinces [43]. Moreover, the protein contents of 21 different sources of *P. cyrtonema* ranged from 3.32% to 4.73% [44]. Previous research has shown that nitrogen loss peaks at a slope of 25° and subsequently declines [45]. Thus, the low protein content of Yueyang1 might be attributed to its planting slope of 20–25°. Our study’s results suggest that, compared with ginger (1.82%), yam (1.53%), and potatoes (2.57%) [46], *P. cyrtonema* is a valuable source of plant protein.

The lipid content was generally low with the highest content measured for Yueyang2 at 0.95%, around twice that of Huanggang (0.45%). In comparison, a previous study showed that *P. cyrtonema* from Guizhou province also has a low average lipid content (0.39%), which varies between strains with different characteristics (0.13–0.66%) [19]. In contrast, a relatively high lipid content was reported for ‘Lijing No.1’ (3.68%), an artificially bred variety, and its lipid content was positively correlated with the altitude [20]. The fat contents of the five germplasms included in this study appear to be related to the altitude with both Yueyang germplasms grown at high altitudes showing higher fat levels.

Chizhou possessed the highest ash content, and there was no significant difference in the ash contents of the other four germplasms. These values are comparable with the results of Jiang et al. (2020), who reported that the ash content of *P. cyrtonema* from Zhejiang ranged from 3.5% to 4.25% [47]. The ash levels of the five *P. cyrtonema* were less affected by environmental factors as their contents were relatively stable. Moreover, as the ash content appears to be related to the total food mineral quantity [48], these five germplasms might be good sources of minerals. 

Carbohydrates constituted the dominant nutrients in *Polygonatum*. The carbohydrate content of *P. cyrtonema* ranged from 23.27% for Yueyang1 to 34.11% for Huanggang, which is similar to the results of a previous study on four types of *P. cyrtonema* from Anhui (23.1–30.9%) [49]. In addition, a wide variation in the carbohydrate content has been demonstrated for another *Polygonatum* species, *P. sibiricum*, ranging from 19.07% to 36.87% [50]. There is a large polysaccharide content in *Polygonatum* rhizomes, making this a crucial quality indicator for this product. The polysaccharide content of the five germplasms varied significantly, ranging from 16.92% to 28.48% (Table 2), indicating that polysaccharides are major contributors to the carbohydrate content. A large variation in the polysaccharide content (6.24–25.30%) was also detected in different *Polygonatum* species collected from 14 provinces and municipalities [51]. Zeng et al. (2021) reported that the *Huangjing* polysaccharide content was significantly negatively correlated with altitude [20], and a similar result was shown for *Ligusticum sinensis’ Chuanxiong’* [52]. Thus, the relatively higher polysaccharide contents of Huanggang, Chizhou, and Qingyang might result from their lower altitudes.

Fructose, glucose, and sucrose are important carbohydrates that contribute to food sweetness [53]. Their contents were significantly different in the five germplasms, with fructose (1.06% to 4.88%) being the most abundant and glucose (0.35% to 0.66%) being the least abundant. Similar results were also reported by Chang et al. (2016) [54] and Jin et al. (2018) [55] for *P. cyrtonema*. In addition, a higher level of fructose relative to glucose and sucrose was also found in *P. cyrtonema* and *P. sibiricum* [56]. Thus, we suggest that fructose is the main contributor to sweetness in *Polygonatum*.

The starch concentration ranged from 7.35% to 10.14%, with an average percentage of 8.52%. This is comparable to the results of Zhang et al. (2022) [18], who reported a starch content of 10.73% as evaluated via acid hydrolysis in *P. cyrtonema*. A lower starch content (2.19% to 6.29%) was also demonstrated in *P. cyrtonema* by Guizhou and Sichuan using the starch content test kit (BC0705) [43]. Si and Zhu (2021) [3] concluded that *Polygonati rhizoma* lacks starch because no blue dots were detected in *P.cyrtonema* rhizome tissue using iodine staining. Although its starch content is still under debate, there is no doubt that *P. cyrtonema* is low in starch compared to yam [57] and cassava [58].

### 3.2. Dietary Fiber Composition of P. cyrtonema

Dietary fiber (DF) is a non-starch polysaccharide that has benefits for the human body, such as preventing and alleviating type 2 diabetes, cardiovascular disease, and colon cancer [59]. In terms of its solubility, DF is classified as soluble (SDF) or insoluble dietary fiber (IDF). SDF is composed of pectin, oligosaccharides, resistant starch, and certain indigestible polysaccharides, while lignin, cellulose, and hemicellulose compose IDF [60,61]. 

Pectin and resistant starch are the main SDF components in *P. cyrtonema*. The pectin substances are the predominant polymeric components of the cell walls, accounting for approximately 37–54% of the total cell wall polysaccharides [62,63]. In the current study, the pectin content varied significantly among the five germplasms, ranging from 0.45% to 1.20% (Table 3). This is lower than the content measured for *Crataegus pinnatifida* Bunge (3–4%) in a previous study [64]. Resistant starch (RS) is a form of SDF that exhibits many potential health benefits [65]. The RS level (1.90–1.58%) detected in *P. cyrtonema* is slightly higher than that in wheat (0.55–0.85%) [66] and lower than that in cassava (2.8–4.1%) [67]. The 1-kestose, nystose, and 1F-fructofuranosylnystose contents of *P. cyrtonema* were measured as 0.21–0.36%, 0.11–0.23%, and 0.12–0.22%, respectively. In comparison, Jin et al. (2018) [55] reported that there was 0.3–1.15% 1-kestose in the fresh rhizomes of *P. cyrtonema*. There were significant differences in the 1-kestose (0.045–0.061%) and nystose (0.004–0.015%) concentrations of garlic at different altitudes [68]. Yueyang1, with a relatively large slope, had the highest contents of RS and oligofructose. These results indicate that the SDF contents were possibly associated with environmental factors, and the planting slope of 20–25° might be conducive to the accumulation of SDF.

Hemicellulose, representing an average concentration of 5.87%, was the primary component of the IDF, accounting for 58.52% of the total IDF. The lignin content was highest for Yueyang1 at 2.64%, and the content of the other germplasms ranged from 2.03% to 2.28%. Therefore, both hemicellulose and lignin were relatively stable among different *P. cyrtonema* sources. The cellulose content ranged from 1.33% in Huanggang to 2.75% in Chizhou. A significant difference in the cellulose content was also reported in 16 *P. kingianum* resources from Yunnan province, with a range of 1.6% to 11% [69]. The concentration of cellulose was related to the varieties and growth stages in quinoa [70]. As a result, cellulose content may be mainly controlled by the genotype. 

Finally, the TDF content was highest in Huanggang and Chizhou, 14.24% and 13.81%, respectively (Table 3). Nevertheless, Yueyang1, with the highest fructooligosaccharide and resistant starch contents, showed the lowest level of TDF, which might be due to the determination method employed. The TDF content measured by the enzyme weight method did not contain low-molecular-weight SDF, such as oligosaccharides and resistant starch [35]. Shen et al. (2020) [71] reported that the crude fiber concentrations of *P. cyrtonema* and *Polygonatum filipes* measured by Van Soest’s washing method ranged from 0.05% to 0.08%, while a fiber content of 33.1% was detected in *P. verticillatum* rhizomes, as determined by the weight loss form the crucible after heating [21]. Regardless of the sources, the determination methods used might account for the significant differences in the *Polygonatum* fiber contents measured in previous studies. Moreover, the TDF content has been shown to be higher in lower-altitude regions, and the SDF content is more susceptible to agricultural conditions than the IDF and TDF contents, a result that has also been reported for potatoes [72]. Furthermore, the average TDF content of the five *P. cyrtonema* germplasms (11.70%) was comparable to that of yam (12.09%) [73] and higher than those of root vegetables, such as potatoes, cassava [74], carrots, beetroot, radish, and horseradish [75]. Therefore, these results suggest that *P. cyrtonema* is high in dietary fiber and meets the dietary fiber requirement of ≥6% specified in the Chinese National Standard GB 28050-2011 [76] and Regulation (EC) No 1924/2006 of the European Parliament and the Council [77].

### 3.3. The Mineral Composition of P. cyrtonema

The macromineral contents of the five different germplasms are shown in Table 4. K was the most abundant mineral (495.95–684.38 mg/100 g), followed by Ca (195.08–282.08 mg/100 g). Both K and Ca were found to be the major minerals in ‘Lijing No.1’, with average contents of 464.44 mg/100 g and 359.98 mg/100 g, respectively [20]. Huanggang had the highest concentrations of K and Ca, whereas Qingyang had the lowest. Accordingly, a wide range of K values (122.52–998.66 mg/100 g) have been identified in *P. sibiricum* germplasms from ten provinces [38]. Huang et al. (2004) [22] reported that *Polygonatum* had the greatest Ca concentration (122.54–469.03 mg/100 g), while K was not evaluated.

The Mg contents also varied significantly among the five germplasms, with the highest value found in Yueyang2 (100.37 mg/100 g), while the lowest was obtained in Huanggang (68.23 mg/100 g). Furthermore, the amounts of Mg in the five germplasms were higher than those reported in previous studies for *P. odoratum* and *P. sibiricum* [38]. Zhang et al. (2022) [18] also revealed that *P. cyrtonema* showed the highest Mg content compared to *P. kingianum*, *P. odoratum*, and *P. sibiricum*.

The micromineral contents of *P. cyrtonema* also differed significantly among the five germplasms (Table 4). The highest contents of Fe and Al were found in Chizhou, while the highest Mn content was detected in Yueyang2. The Na content was significantly higher in Yueyang1 than in the other four germplasms, while the Mn and Zn contents were the lowest. The Fe, Al, and Zn contents of the five germplasms were higher than those of *P. sibiricum* from ten different areas, while the Na and Mn contents of the five germplasms were comparable [38]. As shown in Table 4, the average contents of Fe, Mn, Na, Zn, and Cu were much higher than those found for ginger root, yam, and potatoes [46]. Moreover, the Fe and Mn contents in 10 g of powder from the five germplasms accounted for relatively high proportions of the recommended daily intake (RNI) and the daily appropriate intake (AI), which are 14.84–49.48% and 7.49–30.84%, respectively [78]. Additionally, Zhang et al. (2022) [18] reported that *P. cyrtonema* has significant advantages in terms of supplementing mineral elements, especially Fe and Mn. 

The Cu, Cd, and Pb contents of *P. cyrtonema* were found to be 0.54–0.68 mg/100 g, 0.03–0.06 mg/100 g, and 0.05–0.07 mg/100 g, respectively, which meet the *Chinese Pharmacopoeia’s* standards limits [1]. In comparison, the Cu, Cd, and Pb concentrations in *P. odoratum* and *P. sibiricum* were found to be 0.08–0.47 mg/100 g, 0.001–0.12 mg/100 g, and 0.01–0.29 mg/100 g, respectively [38]. 

The total mineral content of Qingyang was the lowest among the five resources, which is consistent with the ash content results. Mineral nutrients in plants fluctuate greatly due to environmental factors such as the weather, climate, and soil texture types [79,80,81]. All five germplasms planted in Yuhang with similar soil and climatic conditions showed significant differences in the mineral concentration, indicating that this difference might be associated with their original sources. To summarize these results, the five *P. cyrtonema* germplasms are characterized by different mineralogical elements and are also good sources of K, Ca, Mg, Mn, and Fe.

### 3.4. The Vitamin Composition of P. cyrtonema

Vitamins are essential for the proper functioning of organisms and are required in small amounts in the diet because they cannot be synthesized by the body [82]. Vitamin C is an essential micronutrient with a robust antioxidant capacity that facilitates cholesterol metabolism in the liver [83]. The vitamin C content identified in the studied germplasms ranged from 60.49 mg/100 g to 149.86 mg/100 g, with the content in Huanggang being significantly higher than those in the other germplasms (Table 5). In comparison, Khan et al. (2012) [21] reported a higher vitamin C content in *P. verticillatum* (170 mg/100 g). According to Zhang et al. (2022), there is an 82% coefficient of variation in the vitamin C levels of the four *Polygonatum* species [18]. Previous research has also shown that, in potato, vitamin C is the nutrient that differs the most among genotypes and depending on the agricultural conditions [72]. Given that Huanggang was grown in the same environment as Chizhou and Qingyang, these growing conditions appear to be conducive to the accumulation of vitamin C for Huanggang. In addition, *P. cyrtonema* might be a good source of vitamin C, with levels comparable to those of kiwifruit [84] and citrus [85].

Carotenoids are natural fat-soluble pigments found in plants, and their provitamin A and antioxidant activities benefit human health [86]. Thiamine and riboflavin are engaged in a variety of crucial metabolic pathways in the body [82]. Previous research has shown that the vitamin A, thiamin, and riboflavin contents vary among different species of *Polygonatum* [18]. In the current study, the carotenoid, thiamine, and riboflavin levels of *P. cyrtonema* were found to be 0.20–0.61 mg/100 g, 0.06–0.12 mg/100 g, and 0.15–0.41 mg/100 g, respectively (Table 5), with the highest levels of carotenoids and riboflavin found in Huanggang. The samples from Yueyang had the lowest carotenoid contents, a result that was likely influenced by the topography and altitude. Furthermore, the carotenoid content of *P. cyrtonema* was comparable to persimmon fruits, with a range of 0.41–0.76 mg/100 g [87]. As shown in Table 5, the thiamine concentration of *P. cyrtonema* was comparable to yam but was higher than that of ginger and potatoes. In addition, riboflavin was found to be abundant in *P. cyrtonema* compared to ginger, yams, and potatoes.

### 3.5. The Amino Acid Composition of P. cyrtonema

Twenty-four amino acids were detected in the five germplasms, including six essential amino acids (EAAs, Table 6). Val (0.27–0.59 mg/g) and Lys (0.25–0.57 mg/g) were the dominant EAA, followed by Ile, Leu, Thr, and Phe. Zhang et al. (2020) [88] identified seven EAAs in *P. cyrtonema* and *P. filipes*. Moreover, eight EAAs were found in several *Polygonatum* species, including *P. cyrtonema*, *P. odoratum*, and *P. sibiricum*, while *P. kingianum* was found to contain seven EAAs [18]. Nonessential amino acids (NEAAs) accounted for the majority of the amino acids in *P. cyrtonema*, with Asn (13.73–20.47 mg/g) and Arg (2.50–12.89 mg/g) accounting for 77.14–82.66% of the total amino acids (TAAs). It was reported that flavor-inducing amino acids (Asp and Glu) and aromatic amino acids (Tyr and Arg) are relatively abundant in *P. cyrtonema* [88]. Chen et al. (2021) [89] showed that Glu and Arg are the most abundant amino acids in ‘Lijing No.1’. In addition, Cit, Thr, Asp, and Asn were found to be abundant in five different sources of *P. cyrtonema* and *P. sibiricum* [56]. The highest TAA content was detected in Yueyang2 (40.36 mg/g), whereas the lowest was measured for Yueyang1 (20.35 mg/g). Fang et al. (2018) reported that the TAA levels varied significantly in *P. cyrtonema* from 15 different origins (33.1–69.7 mg/g), and *P. cyrtonema* with a high amino acid content was observed in areas with extensive precipitation [90]. In addition, consistent with the protein content, the lowest TAA content was detected in Yueyang1. It is suggested that the large planting slope of Yueyang1 impeded soil moisture and nutrient retention as well as amino acid accumulation in the rhizomes.

### 3.6. Correlation Analysis (CA)

Correlation analysis is a promising approach to understand the complex relationships between various food qualities. In this study, 35 nutritional parameters were used as variables to carry out the statistical analysis. The correlations between the nutritional components of *P. cyrtonema* are shown in Figure 2. There was a strong positive correlation between carbohydrates and polysaccharides because polysaccharides were the most abundant carbohydrates. Moreover, the carbohydrate components had significant correlations with dietary fiber. For example, polysaccharides were negatively correlated with 1-kestose and lignin, sucrose and pectin were significantly positively correlated, and both fructose and glucose were significantly positively associated with hemicellulose. Furthermore, carbohydrates and their components were significantly correlated with vitamins, lipids, and minerals. For instance, carbohydrates had a significantly positive correlation with carotenoids, glucose and lipids were negatively correlated, sucrose was significantly positively correlated with vitamin C, riboflavin, and Pb, and in particular, polysaccharides were positively correlated with Zn, both of which have the potential to reduce the prevalence of prediabetes [91,92]. 

Strong correlations were also found between the dietary fiber fractions. For example, there were significant positive relationships between lignin and 1-kestose, between resistant starch and 1-kestose, and between resistant starch and 1F-fructofuranosylnystose. In addition, the dietary fiber fractions showed significant correlations with vitamins, minerals, TAAs, and proteins. For example, 1F-fructofuranosylnystose was negatively correlated with Mn, proteins, and TAAs; resistant starch, lignin, and 1-kestose exhibited significant negative correlations with Zn; and pectin had significant positive correlations with vitamin C, Pb, and Ca. Our results are in accordance with the close relationship of pectin with Ca, as pectin is a major component of cell walls [93] and must be bound to Ca in order to maintain the structural stability of plant cells [94].

There was a significant positive correlation between protein and TAAs, which has also been reported for potato [95] and yam [96]. TAAs also showed a significant positive relationship with the Mn content. A previous study also displayed a significant positive relationship in walnuts between some minerals and amino acids [97]. Overall, there were complex correlation relationships between the nutrients in *P. cyrtonema*, especially the correlations between the carbohydrate component and dietary fiber fractions.

### 3.7. Principal Component Analysis (PCA)

To gain a more comprehensive understanding of the variation between the *P. cyrtonema* from five different areas, PCA was performed on 35 nutritional parameters. There were four principal components with eigenvalues greater than one, which explained 100.00% of the total variance in all *P. cyrtonema* nutrients (Table 7). A loading plot of the first two PCs and a score plot with the five germplasms are illustrated in Figure 2. The first principal component (PC1) explained 36.29% of the total variance. Polysaccharides and Zn were shown to be the most influential components in the positive region of PC1, and resistant starch was identified as the most influential component in the negative region. PC2 explained 34.45% of the total change and reflected the comprehensive indexes of riboflavin, hemicellulose, glucose, and Mg. Riboflavin, hemicellulose, and glucose were shown to have positive effects on PC2, whereas Mg showed a negative effect. PC3 accounted for 16.44% of the variance and was defined by nystose, Fe, and Al (Table 7). PC4 explained 12.82% of the variance and was represented by ash and Ca (Table 7). Therefore, the nutrient components, including polysaccharides, resistant starch, hemicellulose, glucose, nystose, riboflavin, Zn, Mg, Fe, Al, Ca, and ash, are important variables that can be used to distinguish these five germplasms.

The scoring point positions of the five germplasms were separated in the regions of PC1 and PC2, which can be explained by the fact that the five germplasms present different nutritional characteristics (Figure 3). For example, the highest levels and loading arrow directions of carbohydrates, vitamin C, pectin, TDF, K, and Ca were found in Huanggang. Yueyang1 had the lowest protein and TAA levels but the highest fructose and resistant starch contents, and Yueyang2 was characterized by the highest Mn and protein levels. 

Overall, it appears to be feasible to classify the *P. cyrtonema* accessions according to their nutritional profiles. Considering the abundance, variation, importance, and relevance of the nutrients, the components of polysaccharides, TDF, pectin, resistant starch, protein, fructose, vitamin C, Fe, Ca, and ash were identified as the main potential indicators to evaluate the nutritional quality of *P. cyrtonema*. Furthermore, the comprehensive nutritional quality scores of *P. cyrtonema* were obtained by multiplying the scores of the four principal components by the contribution ratio of each principal component, which acted as the weight. Consequently, the ranking of the comprehensive scores reflects the nutritional quality of *P. cyrtonema*, with Huanggang possessing the highest value (Table 8).

## 4. Conclusions

In this study, the nutritional compositions of five *P. cyrtonema* germplasms were systematically evaluated and analyzed. There were significant differences in the proximate, carbohydrate, and dietary fiber contents as well as the mineral, vitamin, and amino acid compositions among the different germplasms. Our study demonstrates that the five artificially cultivated *P. cyrtonema* are excellent sources of protein, carbohydrates, dietary fiber, vitamin C, and minerals such as Ca, Fe, and Mn in comparison with ginger, yam, and potatoes. A correlation analysis revealed significant correlations of the carbohydrate components and dietary fiber fractions with other nutrients; for example, there was a positive correlation between polysaccharides and Zn and between pectin and Ca. Moreover, the PCA analysis showed that the five germplasms were distinguished successfully in terms of their nutritional characteristics, with Huanggang possessing the highest nutritional quality. Finally, ten characteristic indicators were selected from the comprehensive nutrient analysis of the five *P. cyrtonema* germplasms. In conclusion, this study demonstrates that *P. cyrtonema* is a nutritious food, and the obtained nutritional indicators can provide valuable guidance for the identification and evaluation of *Polygonatum*.

## Figures and Tables

**Figure 1 foods-13-00815-f001:**
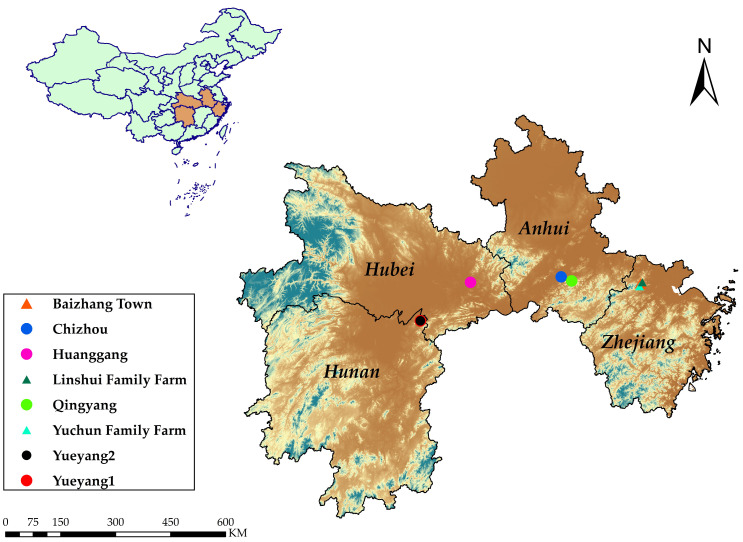
Geographical map of the provenance and planting location of the five *P. cyrtonema*.

**Figure 2 foods-13-00815-f002:**
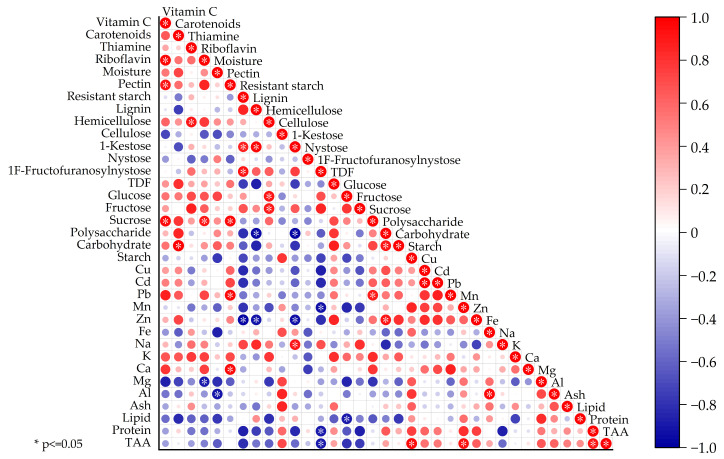
Correlation analysis of nutritional components in five *P. cyrtonema*.

**Figure 3 foods-13-00815-f003:**
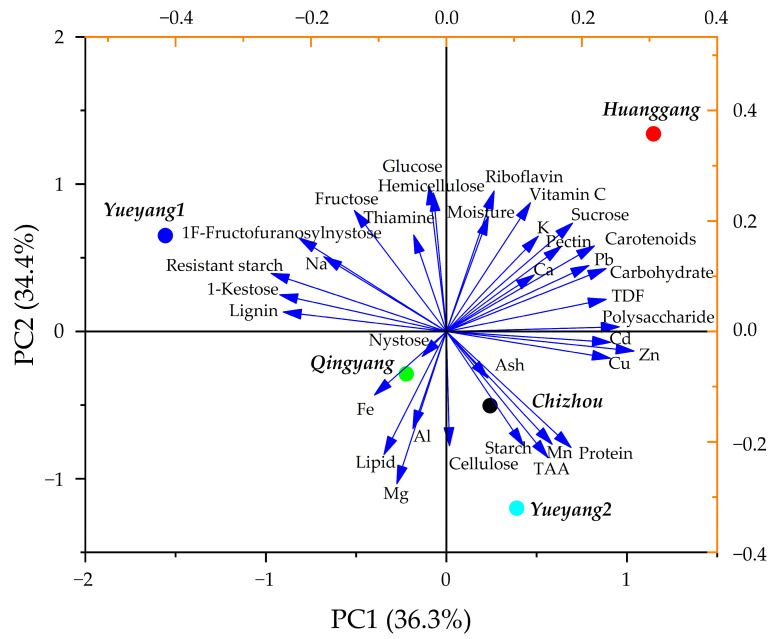
Loading and score plot of PCA analysis for the nutrients of five *P. cyrtonema*.

**Table 1 foods-13-00815-t001:** Specific information on the planting location of the five *P. cyrtonema*.

Sample Name	Voucher Specimen Number	Location	Latitude (N)	Longitude (E)	Altitude (m)	Slope (°)	Average Temperatures (°C)	Annual Rainfall (mm)
Huanggang	ZSTU211101	Common garden	30°29′08″	119°45′37″	120	5–10	17.22	2296.62
Chizhou	ZSTU211102							
Qingyang	ZSTU211103							
Yueyang1	ZSTU211104	Yuchun family farm	30°24′56″	119°43′43″	180	20–25		
Yueyang2	ZSTU211105	Linshui family farm	30°30′41″	119°48′41″	235	10–15		

Note: Data on average temperature and annual rainfall in 2021 were obtained from the National Centers for Environmental Information (https://www.ngdc.noaa.gov/) (accessed on 3 December 2023).

**Table 2 foods-13-00815-t002:** Proximate and carbohydrate composition of the *P. cyrtonema* from different origins (%).

Nutrient	Huanggang	Chizhou	Qingyang	Yueyang1	Yueyang2
Moisture	5.39 ± 0.11 ^a^	4.59 ± 0.09 ^c^	5.27 ± 0.68 ^ab^	4.74 ± 0.10 ^bc^	4.28 ± 0.12 ^c^
Protein	4.25 ± 0.19 ^c^	5.23 ± 0.17 ^a^	4.78 ± 0.12 ^b^	2.96 ± 0.00 ^d^	5.40 ± 0.06 ^a^
Lipid	0.45 ± 0.09 ^c^	0.60 ± 0.05 ^bc^	0.68 ± 0.03 ^b^	0.72 ± 0.13 ^b^	0.95 ± 0.09 ^a^
Ash	3.01 ± 0.07 ^b^	4.24 ± 0.05 ^a^	2.88 ± 0.07 ^b^	2.90 ± 0.04 ^b^	3.03 ± 0.08 ^b^
Carbohydrate	34.29 ± 1.05 ^a^	29.96 ± 0.35 ^b^	28.88 ± 0.32 ^b^	23.25 ± 0.45 ^d^	25.90 ± 0.40 ^c^
Polysaccharide	28.48 ± 0.99 ^a^	25.59 ± 0.43 ^b^	25.99 ± 0.30 ^b^	16.92 ± 0.43 ^d^	23.59 ± 0.40 ^c^
Glucose	0.66 ± 0.05 ^a^	0.55 ± 0.03 ^b^	0.54 ± 0.02 ^b^	0.61 ± 0.01 ^ab^	0.35 ± 0.01 ^c^
Fructose	3.44 ± 0.05 ^b^	2.91 ± 0.18 ^c^	1.69 ± 0.02 ^d^	4.88 ± 0.03 ^a^	1.06 ± 0.01 ^e^
Sucrose	1.72 ± 0.05 ^a^	0.92 ± 0.05 ^b^	0.66 ± 0.01 ^c^	0.84 ± 0.03 ^b^	0.90 ± 0.02 ^b^
Starch	7.83 ± 0.33 ^b^	9.90 ± 0.33 ^a^	7.36 ± 0.21 ^b^	7.35 ± 0.52 ^b^	10.14 ± 0.35 ^a^

Data are expressed as the mean ± SD (*n* = 3). Means with different superscripts within a row are significantly different from each other (*p* < 0.05).

**Table 3 foods-13-00815-t003:** Fiber composition of the *P. cyrtonema* from different origins (%).

Fiber		Huanggang	Chizhou	Qingyang	Yueyang1	Yueyang2
SDF	Pectin	1.20 ± 0.07 ^a^	0.63 ± 0.02 ^c^	0.45 ± 0.04 ^d^	0.69 ± 0.05 ^c^	0.84 ± 0.06 ^b^
	Resistant starch	1.61 ± 0.02 ^c^	1.61 ± 0.03 ^c^	1.72 ± 0.08 ^b^	1.90 ± 0.04 ^a^	1.58 ± 0.02 ^c^
	1-Kestose	0.21 ± 0.01 ^b^	0.21 ± 0.04 ^b^	0.22 ± 0.01 ^b^	0.36 ± 0.02 ^a^	0.24 ± 0.04 ^b^
	Nystose	0.18 ± 0.02 ^b^	0.18 ± 0.04 ^b^	0.15 ± 0.01 ^bc^	0.23 ± 0.01 ^a^	0.11 ± 0.00 ^c^
	1F-Fructofuranosylnystose	0.16 ± 0.01 ^b^	0.16 ± 0.03 ^b^	0.17 ± 0.00 ^b^	0.22 ± 0.01 ^a^	0.12 ± 0.01 ^c^
IDF	Hemicellulose	5.99 ± 0.10 ^a^	5.91 ± 0.12 ^ab^	5.78 ± 0.16 ^ab^	5.96 ± 0.07 ^ab^	5.70 ± 0.05 ^b^
	Cellulose	1.33 ± 0.14 ^c^	2.75 ± 0.30 ^a^	1.75 ± 0.22 ^bc^	1.67 ± 0.14 ^bc^	2.11 ± 0.25 ^b^
	Lignin	2.08 ± 0.00 ^b^	2.03 ± 0.21 ^b^	2.19 ± 0.05 ^b^	2.64 ± 0.13 ^a^	2.28 ± 0.05 ^b^
TDF		14.24 ± 1.38 ^a^	13.84 ± 0.18 ^a^	10.39 ± 0.14 ^b^	9.05 ± 0.25 ^c^	11.00 ± 0.22 ^b^

Data are expressed as the mean ± SD (*n* = 3). Means with different superscripts within a row are significantly different from each other (*p* < 0.05).

**Table 4 foods-13-00815-t004:** Mineral composition of the *P. cyrtonema* from different origins (mg/100 g).

Mineral	Huanggang	Chizhou	Qingyang	Yueyang1	Yueyang2	Ginger Root, Raw ^1^	Yam, Raw ^1^	Potatoes, Raw, Skin ^1^
K	684.38 ± 25.57 ^a^	639.35 ± 49.60 ^a^	495.95 ± 9.71 ^c^	571.51 ± 11.04 ^b^	534.29 ± 10.43 ^bc^	415.00	816.00	413.00
Ca	282.08 ± 13.37 ^a^	223.17 ± 20.33 ^c^	195.08 ± 13.20 ^d^	240.33 ± 6.15 ^bc^	264.67 ± 2.50 ^b^	16.00	17.00	30.00
Mg	68.23 ± 2.16 ^d^	93.78 ± 8.33 ^ab^	89.07 ± 1.99 ^bc^	85.36 ± 2.10 ^c^	100.37 ± 1.69 ^a^	43.00	21.00	23.00
Fe	32.30 ± 0.98 ^c^	59.37 ± 4.63 ^a^	29.68 ± 2.47 ^c^	55.37 ± 1.61 ^ab^	52.48 ± 1.84 ^b^	0.60	0.54	3.24
Al	39.16 ± 0.93 ^d^	74.03 ± 1.94 ^a^	40.02 ± 3.72 ^d^	58.48 ± 4.53 ^c^	66.79 ± 1.03 ^b^	NI	NI	NI
Na	16.52 ± 2.58 ^b^	11.06 ± 1.75 ^c^	5.11 ± 2.36 ^d^	36.09 ± 0.74 ^a^	13.82 ± 0.80 ^bc^	13	9.00	10.00
Mn	6.85 ± 0.56 ^c^	8.28 ± 0.57 ^b^	5.83 ± 0.38 ^d^	3.37 ± 0.25 ^e^	13.88 ± 0.43 ^a^	0.23	0.40	0.60
Zn	6.97 ± 0.49 ^a^	6.32 ± 0.85 ^ab^	5.53 ± 0.36 ^b^	3.56 ± 0.68 ^c^	6.28 ± 0.21 ^ab^	0.34	0.24	0.35
Cu	0.67 ± 0.02 ^a^	0.59 ± 0.09 ^ab^	0.61 ± 0.03 ^ab^	0.54 ± 0.02 ^b^	0.68 ± 0.02 ^a^	0.23	0.18	0.42
Cd	0.06 ± 0.00 ^a^	0.04 ± 0.00 ^b^	0.04 ± 0.00 ^b^	0.03 ± 0.00 ^c^	0.06 ± 0.00 ^a^	NI	NI	NI
Pb	0.07 ± 0.00 ^a^	0.05 ± 0.01 ^b^	0.05 ± 0.01 ^b^	0.05 ± 0.00 ^b^	0.06 ± 0.00 ^a^	NI	NI	NI

Data are expressed as the mean ± SD (*n* = 3). Means with different superscripts within a row are significantly different from each other (*p* < 0.05). ^1^ Adapted from the USDA nutrient database (U.S. Department of Agriculture, Agricultural Research Service, 2019) [46]. NI = not informed.

**Table 5 foods-13-00815-t005:** Vitamin composition of the *P. cyrtonema* from different origins (mg/100 g).

Vitamin	Huanggang	Chizhou	Qingyang	Yueyang1	Yueyang2	Ginger Root, Raw ^1^	Yam, Raw ^1^	Potatoes, Raw, Skin ^1^
Vitamin C	149.86 ± 6.52 ^a^	60.49 ± 2.31 ^d^	60.71 ± 2.72 ^d^	88.09 ± 3.44 ^b^	75.57 ± 1.26 ^c^	5	17.1	11.4
Carotenoids	0.61 ± 0.03 ^a^	0.39 ± 0.04 ^b^	0.38 ± 0.05 ^b^	0.20 ± 0.03 ^c^	0.25 ± 0.03 ^c^	NI	NI	NI
Thiamine	0.11 ± 0.01 ^a^	0.12 ± 0.02 ^a^	0.06 ± 0.01 ^b^	0.12 ± 0.01 ^a^	0.06 ± 0.01 ^b^	0.025	0.112	0.021
Riboflavin	0.41 ± 0.00 ^a^	0.19 ± 0.03 ^c^	0.15 ± 0.00 ^d^	0.29 ± 0.02 ^b^	0.19 ± 0.01 ^c^	0.034	0.032	0.038

Data are expressed as the mean ± SD (*n* = 3). Means with different superscripts within a row are significantly different from each other (*p* < 0.05). ^1^ Adapted from the USDA nutrient database (U.S. Department of Agriculture, Agricultural Research Service, 2019) [46]. NI = not informed.

**Table 6 foods-13-00815-t006:** Amino acid composition of the *P. cyrtonema* from different origins (dried weight: mg/g).

Amino Acid Composition		Huanggang	Chizhou	Qingyang	Yueyang1	Yueyang2
EAA	Valine (Val)	0.41 ± 0.02 ^c^	0.51 ± 0.03 ^b^	0.40 ± 0.02 ^c^	0.27 ± 0.01 ^d^	0.59 ± 0.03 ^a^
	Isoleucine (Ile)	0.17 ± 0.01 ^c^	0.22 ± 0.01 ^b^	0.16 ± 0.01 ^c^	0.14 ± 0.01 ^c^	0.32 ± 0.02 ^a^
	Leucine (Leu)	0.19 ± 0.01 ^c^	0.23 ± 0.01 ^b^	0.23 ± 0.01 ^b^	0.15 ± 0.00 ^d^	0.37 ± 0.01 ^a^
	Phenylalanine (Phe)	0.12 ± 0.01 ^d^	0.23 ± 0.01 ^b^	0.18 ± 0.01 ^c^	0.06 ± 0.00 ^e^	0.27 ± 0.02 ^a^
	Lysine (Lys)	0.32 ± 0.02 ^c^	0.50 ± 0.03 ^b^	0.46 ± 0.03 ^b^	0.25 ± 0.01 ^d^	0.57 ± 0.03 ^a^
	Threonine (Thr)	0.29 ± 0.02 ^a^	0.32 ± 0.02 ^a^	0.20 ± 0.01 ^b^	0.17 ± 0.01 ^b^	0.33 ± 0.02 ^a^
NEAA	Phosphoserine (P-Ser)	0.20 ± 0.01 ^a^	0.20 ± 0.01 ^a^	0.20 ± 0.01 ^a^	0.21 ± 0.01 ^a^	0.19 ± 0.01 ^a^
	Aspartic acid (Asp)	0.52 ± 0.04 ^a^	0.40 ± 0.03 ^b^	0.29 ± 0.02 ^c^	0.21 ± 0.02 ^cd^	0.27 ± 0.02 ^d^
	Serine (Ser)	0.63 ± 0.02 ^b^	0.64 ± 0.02 ^b^	0.35 ± 0.01 ^c^	0.33 ± 0.01 ^c^	0.83 ± 0.03 ^a^
	Asparagine (Asn)	11.99 ± 0.02 ^e^	17.8 ± 0.03 ^b^	15.4 ± 0.03 ^c^	13.73 ± 0.03 ^d^	20.47 ± 0.04 ^a^
	Proline (Pro)	0.61 ± 0.03 ^a^	0.65 ± 0.03 ^a^	0.41 ± 0.02 ^c^	0.33 ± 0.01 ^d^	0.53 ± 0.02 ^b^
	Glycine (Gly)	0.11 ± 0.00 ^a^	0.11 ± 0.00 ^a^	0.07 ± 0.00 ^c^	0.06 ± 0.00 ^d^	0.10 ± 0.00 ^b^
	Alanine (Ala)	0.64 ± 0.02 ^b^	0.88 ± 0.03 ^a^	0.49 ± 0.01 ^c^	0.42 ± 0.01 ^d^	0.63 ± 0.02 ^b^
	Cystine (Cys-Cys)	0.11 ± 0.01 ^c^	0.17 ± 0.01 ^ab^	0.16 ± 0.01 ^b^	0.07 ± 0.00 ^d^	0.19 ± 0.01 ^a^
	Tyrosine (Tyr)	0.18 ± 0.01 ^a^	0.19 ± 0.01 ^a^	0.17 ± 0.01 ^a^	0.05 ± 0.00 ^c^	0.08 ± 0.01 ^b^
	γ-aminobutyric acid (GABA)	0.98 ± 0.00 ^d^	1.38 ± 0.00 ^a^	1.02 ± 0.00 ^c^	0.64 ± 0.00 ^e^	1.07 ± 0.00 ^b^
	L-ornithine (Orn)	0.05 ± 0.00 ^e^	0.13 ± 0.00 ^c^	0.10 ± 0.00 ^d^	0.24 ± 0.00 ^a^	0.16 ± 0.00 ^b^
	Arginine (Arg)	8.49 ± 0.30 ^c^	11.11 ± 0.39 ^b^	6.18 ± 0.22 ^d^	2.50 ± 0.09 ^e^	12.89 ± 0.45 ^a^
	Taurine (Tau)	0.03 ± 0.00 ^a^	0.03 ± 0.00 ^a^	ND	0.03 ± 0.00 ^a^	0.03 ± 0.00 ^a^
	Phosphoethanolamine (PEA)	0.15 ± 0.00 ^b^	0.18 ± 0.00 ^a^	ND	ND	ND
	Glutamic acid (Glu)	0.10 ± 0.00 ^d^	0.18 ± 0.00 ^b^	0.31 ± 0.01 ^a^	0.16 ± 0.00 ^c^	ND
	Citrulline (Cit)	0.08 ± 0.00 ^d^	0.19 ± 0.00 ^b^	ND	0.20 ± 0.00 ^a^	0.17 ± 0.00 ^c^
	Histidine (His)	0.18 ± 0.01	ND	ND	ND	ND
	1-methylhistidine (1Mhis)	ND	0.30 ± 0.00 ^a^	0.24 ± 0.00 ^b^	0.13 ± 0.00 ^c^	0.30 ± 0.00 ^a^
EAA		1.50 ± 0.09 ^c^	2.01 ± 0.11 ^b^	1.63 ± 0.09 ^c^	1.04 ± 0.04 ^d^	2.45 ± 0.13 ^a^
TAA		26.55 ± 0.57 ^c^	36.55 ± 0.68 ^b^	27.02 ± 0.44 ^c^	20.35 ± 0.23 ^d^	40.36 ± 0.74 ^a^

Data are expressed as the mean ± SD (*n* = 3). Means with different superscripts within a row are significantly different from each other (*p* < 0.05). ND = not detected.

**Table 7 foods-13-00815-t007:** Factor loading, eigenvalue, and contribution ratio of the four principal components.

Nutrient	PC 1	PC 2	PC 3	PC 4
Vitamin C	0.124	0.233	0.063	−0.169
Carotenoids	0.219	0.155	−0.080	0.122
Thiamine	−0.048	0.175	0.265	0.210
Riboflavin	0.070	0.254	0.125	−0.123
Moisture	0.062	0.208	−0.265	0.080
Pectin	0.170	0.154	0.163	−0.208
Resistant starch	−0.259	0.105	−0.046	−0.021
Lignin	−0.241	0.035	0.054	−0.227
Hemicellulose	−0.019	0.251	0.145	0.160
Cellulose	0.004	−0.207	0.172	0.265
1-Kestose	−0.246	0.066	0.112	−0.152
Nystose	−0.036	−0.045	−0.401	0.088
1F-Fructofuranosylnystose	−0.218	0.169	−0.009	0.105
TDF	0.236	0.058	0.116	0.197
Glucose	−0.026	0.263	−0.023	0.187
Fructose	−0.136	0.219	0.165	0.081
Sucrose	0.186	0.196	0.109	−0.076
Polysaccharide	0.255	0.008	−0.134	0.123
Carbohydrate	0.236	0.114	−0.071	0.155
Starch	0.114	−0.207	0.235	0.027
Cu	0.242	−0.048	−0.038	−0.221
Cd	0.242	−0.020	0.027	−0.234
Pb	0.211	0.119	0.054	−0.235
Mn	0.156	−0.204	0.104	−0.167
Zn	0.277	−0.036	0.000	0.046
Fe	−0.106	−0.115	0.343	0.066
Na	−0.181	0.135	0.212	−0.152
K	0.136	0.172	0.237	0.137
Ca	0.130	0.102	0.229	−0.283
Mg	−0.073	−0.276	0.048	0.006
Al	−0.050	−0.176	0.310	0.096
Ash	0.061	−0.084	0.210	0.369
Lipid	−0.092	−0.223	0.029	−0.254
Protein	0.184	−0.210	−0.034	0.081
TAA	0.150	−0.228	0.123	0.007
Eigenvalue	12.70	12.06	5.75	4.49
Contribution ratio (%)	36.29	34.45	16.44	12.82
Cumulative contribution ratio (%)	36.29	70.74	87.18	100.00

**Table 8 foods-13-00815-t008:** Principal component scores and comprehensive scores of *P. cyrtonema* nutritional quality.

Origin	PC 1	PC 2	PC 3	PC 4	Scores	Rankings
Huanggang	1.148	1.341	0.037	−0.288	0.848	1
Chizhou	0.241	−0.504	0.804	1.497	0.238	2
Qingyang	−0.222	−0.289	−1.720	0.328	−0.421	5
Yueyang1	−1.557	0.651	0.494	−0.329	−0.302	3
Yueyang2	0.390	−1.200	0.386	−1.208	−0.363	4

## Data Availability

The original contributions presented in the study are included in the article, further inquiries can be directed to the corresponding author.

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
