# Peer review of "Comprehensive Evaluation of the Nutritional Properties of Different Germplasms of Polygonatum cyrtonema Hua"

_foods, 2024, doi:10.3390/foods13060815_

Round 1

Reviewer 1 Report

Comments and Suggestions for Authors

The current work by Lu et al., was an attempt to comprehensively evaluate the nutrient composition and nutritional quality of Polygonatum cyrtonema Hua. obtained from five different areas in China. The notion of the work and potential implications, i.e., to provide a strong rational and scientific basis for the processing and utilization of the plant sample is indeed worthwhile

 Nonetheless, there are several concerns arising from the manner in which the investigation was performed, results presented and interpreted as well as the write-up in its entirety that warrant further consideration. Firstly, the work offered very little by way of contribution to knowledge or novelty since most of the results were predictable and not different from already published similar works of the same plant sample.

Secondly, the write-up is very poor in several respects. For instance, the Introduction section is very sketchy and provided little information on previous studies related to the broader theme of the investigation. There authors didn't provide a strong enough justification of the need or urgency for the current investigation beside the fact that the number of studies previously reported are few. Why is this subject a research imperative?

The Methods section failed to meet the standards of reproducibility and replicability by failing to provide full details of the step-by-step process of how the protocols were performed. Plant sample authentication was incomplete without details of the specimen voucher number and whether the specimen of. the identified samples were domiciled in reputable national or regional herbarium. What was the geographical coordinates of the different plant collection areas? What was the geographical and climatic characteristic of these locations? What makes these areas different and why were they sele

The Results and discussion section raises some interesting questions. The values obtained for the mineral composition analysis seemed to high to be credible, e.g., K content of 495.95 mg/100 g and 684.38 mg/100 g for samples obtained from Qingyang and Huanggang, respectively. In areas where variations were observed, very little or no explanations were provided for the variability. In all, the work in its current form failed to meet the riquor and standards required for publication. Authors are invited to consider the aforementioned points prior to future re-submission.

Comments on the Quality of English Language

The manuscript needs to be meticulously edited for English language by a professional or native English speaker.

Author Response

To Reviewer #1

The current work by Lu et al., was an attempt to comprehensively evaluate the nutrient composition and nutritional quality of Polygonatum cyrtonema Hua. obtained from five different areas in China. The notion of the work and potential implications, i.e., to provide a strong rational and scientific basis for the processing and utilization of the plant sample is indeed worthwhile.

Nonetheless, there are several concerns arising from the manner in which the investigation was performed, results presented and interpreted as well as the write-up in its entirety that warrant further consideration.

Re: Firstly, the work offered very little by way of contribution to knowledge or novelty since most of the results were predictable and not different from already published similar works of the same plant sample.

Au: Thanks for your suggestion. Polygonatum cyrtonema Hua is mainly consumed as food,but lack of comprehensive nutritional analysis components, which is important to develop new products and further guide consumer consumption. In this study, the total nutrient composition of P. cyrtonema was investigated, and the nutritional quality of the five germplasms after artificial cultivation was evaluated. We demonstrated that P. cyrtonema was a nutritious food, with a low starch content and greater amounts of carbohydrate, protein, Ca, Fe, and vitamin C in comparison with ginger, yam, and potatoes. After analyzed the composition of dietary fiber, P. cyrtonema was found to be high in dietary fiber with pectin and resistant starch being the major soluble components and hemicellulose being the dominant insoluble dietary fiber. The complex correlation relationships between 35 nutritional parameters, especially carbohydrate components and dietary fiber fractions in P. cyrtonema, were elucidated. Furthermore, the nutritional characteristics of the five germplasms and ten characteristic nutritional indicators were obtained by principal component analysis, which would provide valuable references for future development and utilization of Polygonatum.

Re: Secondly, the write-up is very poor in several respects. For instance, the Introduction section is very sketchy and provided little information on previous studies related to the broader theme of the investigation. There authors didn't provide a strong enough justification of the need or urgency for the current investigation beside the fact that the number of studies previously reported are few. Why is this subject a research imperative?

Au: Thanks for your kind suggestions. We rewrote the introduction to provide more information about that Polygonatum was mainly used as a food. With the increasing utilization of Polygonatum in food processing, it is necessary to further explore the nutritional composition of Polygonatum as well as to evaluate the nutritional quality of different germplasm resources. Please see details in the introduction section of the revised manuscript (Lines 32-87).

Re: The Methods section failed to meet the standards of reproducibility and replicability by failing to provide full details of the step-by-step process of how the protocols were performed.

Au: Thank you for reading this manuscript carefully. We checked and supplemented the references and some of the experimental methods with specific steps. For example, the experimental methods for moisture, ash, protein, lipids, and thiamine were entirely based on Chinese national standards. We added specific steps for determining polysaccharides, the composition of dietary fiber, vitamin C, and carotenoid. Please see details in the Methods section of the revised manuscript (Lines 107-240).

Re: Plant sample authentication was incomplete without details of the specimen voucher number and whether the specimen of. the identified samples were domiciled in reputable national or regional herbarium. What was the geographical coordinates of the different plant collection areas? What was the geographical and climatic characteristic of these locations? What makes these areas different and why were they sele.

Au: Thanks for your helpful comment. The plants used in this study were all wild sources and were introduced in Yuhang District, Hangzhou City, Zhejiang Province. Therefore, their growth environment conditions were almost similar. All samples were authenticated by Prof. Zongsuo Liang (Zhejiang Sci-Tech University) and deposited in Key Laboratory of Plant Secondary Metabolism Regulation of Zhejiang Province. After three years of cultivation, each sample was collected for nutritional analysis. The voucher specimen number and specific information on the planting location of the five P. cyrtonema were supplemented and listed in Table 1. Please see the details of the sample source information in the Materials and Methods section of the revised manuscript (Lines 89-106).

Re: The Results and discussion section raises some interesting questions. The values obtained for the mineral composition analysis seemed to high to be credible, e.g., K content of 495.95 mg/100 g and 684.38 mg/100 g for samples obtained from Qingyang and Huanggang, respectively. In areas where variations were observed, very little or no explanations were provided for the variability.

Au: Thanks for your suggestions. We carefully considered the environmental characteristics of five samples and re discussed the specific factors affecting different nutrients contents. Since all samples were planted in similar locations with the same climatic conditions, differences in planting slope and altitude, as well as genetic variations between samples, might be the main factors affecting nutrient composition. Please see details in the Results and Discussion section of the revised manuscript (Lines 242-457).

Re: The manuscript needs to be meticulously edited for English language by a professional or native English speaker.

Au: Thanks for your carefully reading for this manuscript. We sent it to the MDPI Author Services Coordinator to edit for language. All change was highlighted in red color.

In addition, we have corrected the names of two samples from Hunan, changing from "Shaoyang" to "Yueyang". Thank you very much for your valuable comments for further improvement of our manuscript.

Reviewer 2 Report

Comments and Suggestions for Authors

The article “Nutrient composition and nutritional quality evaluation of Polygonatum cyrtonema Hua. from five producing areas” aims to study the proximate content, carbohydrate, dietary fiber, minerals, 11 vitamins, and amino acid composition in P. cyrtomena from five different areas. The article is well written, presents interesting and varied results, which value the fruit under study. In any case, as this is a study that compares fruits from different regions, the discussions should take into account the influence of the geoclimatic characteristics of these regions, and thus improve the discussion and/or formulation of hypotheses regarding the reasons for the significant differences found in the composition of the fruits.

Here are some considerations related to the article:

- Methodology:

The locations where the samples were obtained should be better described (map, characteristics of the location), as temperature, altitude, type of soil could influence the proximate characteristics of the fruits, as discussed in the article, in the case of artificial production.I don't say anything you say:

However, a relatively high lipid content was reported in 'Lijing No.1' (3.68%), an artificially bred variety [15], suggesting that “artificial cultivation” might enhance the fat content of Polygonatum. The lipid content was associated with genotype, “environmental conditions”, and irrigation management, which was also reported in walnuts [29]

For the same reason, the quantification of ash showed a significant difference 4.24±0.05a and 2.88±0.07b. This may be related to the place where the fruit is produced, and is the same for the other parameters. The discussion must take into account the geoclimatic characteristics in the composition of the fruits.

Therefore, it is clear that there are influences from the characteristics of the location. And in this case, which ones? This needs to be discussed, since just focusing on the fact that there are differences does not characterize a sufficiently deep analysis.

-  Dietary fiber composition of P. cyrtonema

The results are well discussed and compared with the literature, but again, it is necessary to associate them with the characteristics of the place where the fruits were harvested.

- The mineral composition of P. cyrtonema

Great discussion and comparison of results, but nothing was associated with the type of soil/environment, something that can influence the concentration of these minerals.

- The vitamin composition of P. cyrtonema

What could cause these significant differences, given that they are fruits of the same species?

- The amino acid composition of P. cyrtonema

As previously asked, at this point, you respond:

Significant variation of TAA was also found in the P. cyrtonema from 15 different areas, ranging from 33.1 mg/g to 69.7 mg/g, which might be “affected by geography, climate, and precipitation”.

The authors should provide a clearer, and not so generic, answer.

Author Response

To Reviewer #2:

The article “Nutrient composition and nutritional quality evaluation of Polygonatum cyrtonema Hua. from five producing areas” aims to study the proximate content, carbohydrate, dietary fiber, minerals, 11 vitamins, and amino acid composition in P. cyrtomena from five different areas. The article is well written, presents interesting and varied results, which value the fruit under study. In any case, as this is a study that compares fruits from different regions, the discussions should take into account the influence of the geoclimatic characteristics of these regions, and thus improve the discussion and/or formulation of hypotheses regarding the reasons for the significant differences found in the composition of the fruits.

Here are some considerations related to the article:

Re: - Methodology:

The locations where the samples were obtained should be better described (map, characteristics of the location), as temperature, altitude, type of soil could influence the proximate characteristics of the fruits, as discussed in the article, in the case of artificial production. I don't say anything you say:

Au: We are grateful for your advice. We provided the geographical and climatic of their origins (Table S1) and the information for cultivation location (Table 1) in the Materials section (Lines 89-106). Since all samples were planted in similar locations with the same climatic conditions, differences in planting slope and altitude, as well as genetic variations between samples, might be the main factors affecting nutrient composition of Polygonatum. We discussed these factors associated with nutritioal componets in the Results and Discussion section of the revised manuscript highlighted with red color.

Re: However, a relatively high lipid content was reported in 'Lijing No.1' (3.68%), an artificially bred variety [15], suggesting that “artificial cultivation” might enhance the fat content of Polygonatum. The lipid content was associated with genotype, “environmental conditions”, and irrigation management, which was also reported in walnuts [29]

Au: Thanks for your helpful comment. “artificial cultivation” should be “artificial selection”. We have rediscussed the factors influencing fat contents of these five germplasms. The fat content of P. cyrtonema might be positively correlated with altitude. Please see details in the revised manuscript (Lines 257-264).

Re: For the same reason, the quantification of ash showed a significant difference 4.24±0.05a and 2.88±0.07b. This may be related to the place where the fruit is produced, and is the same for the other parameters. The discussion must take into account the geoclimatic characteristics in the composition of the fruits.

Au: Thank you for your kind suggestion. The ash content of P. cyrtonema were less affected by environmental factors, as their contents were relatively stable. Please see details in the revised manuscript (Lines 265-271).

Re: Therefore, it is clear that there are influences from the characteristics of the location. And in this case, which ones? This needs to be discussed, since just focusing on the fact that there are differences does not characterize a sufficiently deep analysis.

Au: Thanks for your helpful comment. We carefully considered the environmental characteristics of five samples and rediscussed the specific factors affecting each nutrients content. Please see details in the Results and Discussion section of the revised manuscript highlighted with red color (Lines 242-457).

Re: Dietary fiber composition of P. cyrtonema

The results are well discussed and compared with the literature, but again, it is necessary to associate them with the characteristics of the place where the fruits were harvested.

Au: Thank you for reading this manuscript carefully. We have rediscussed the factors that affect dietary fiber contents of these five germplasms. Resistant starch and oligofructose were possibly associated with environmental factors, and the planting slope of 20°-25° might be conducive to the accumulation of SDF. TDF content might affected by altitude. Please see details in the revised manuscript (Lines 312-356).

Re: The mineral composition of P. cyrtonema

Great discussion and comparison of results, but nothing was associated with the type of soil/environment, something that can influence the concentration of these minerals.

Au: Thank you for your kind suggestion. We have rediscussed the factors influencing the mineral composition of these five germplasms. Considiering their similar growth condition, the difference of mineral composition might associate with its original sources. Please see details in the revised manuscript (Lines 394-401).

Re: The vitamin composition of P. cyrtonema

What could cause these significant differences, given that they are fruits of the same species?

Au: Thank you for your suggestion. We have rediscussed the factors related to the vitamin contents of these five germplasms. Please see details in the revised manuscript (Lines 407-437).

Re: The amino acid composition of P. cyrtonema

As previously asked, at this point, you respond:

Significant variation of TAA was also found in the P. cyrtonema from 15 different areas, ranging from 33.1 mg/g to 69.7 mg/g, which might be “affected by geography, climate, and precipitation”.

The authors should provide a clearer, and not so generic, answer.

Au: Thank you for reading this manuscript carefully. The amino acid contents of these five germplasms might have a correlation with cultivation slope. Please see details in the revised manuscript (Lines 452-457).

In addition, we have corrected the names of two samples from Hunan, changing from "Shaoyang" to "Yueyang". Thank you very much for your valuable comments for further improvement of our manuscript. And we also modified language and highlighted it with red color. Thank you for your positive evaluation and support for the MS.

Reviewer 3 Report

Comments and Suggestions for Authors

The manuscript is written correctly and with a great care. However I have some remarks:

1) The title should be change. Nutrient composition and nutritional quality evaluation together don't sound good.

2) Why did you coose only rhizome for analysis? I know that another parts of the plant also are use as a food and medicine. It could be interesting to compare varied part of the plant in relation to nutrient composition.

3) In my opinion the manuscript should include also the analysis of main active compounds of the plant.

4) The introduction should be rewritten. Add more information about the plant as a food.

5) I couldn't find the novelity of the manuscript.

Author Response

To Reviewer #3:

The manuscript is written correctly and with a great care. However, I have some remarks:

Re: 1) The title should be change. Nutrient composition and nutritional quality evaluation together don't sound good.

Au: We are grateful for your advice. The title was changed to “Comparative analysis of the compositional characterization and nutritional quality of five sources of Polygonatum cyrtonema Hua”.

Re: 2) Why did you choose only rhizome for analysis? I know that other parts of the plant also are used as a food and medicine. It could be interesting to compare varied part of the plant in relation to nutrient composition.

Au: Thank you for your kind suggestion. Currently, the rhizome of Polygonatum was the only medicinal parts specified in the Chinese Pharmacopoeia. Polygonatum rhizomes were mainly consumed as food in China,but lack of comprehensive nutritional analysis components (Lines 53-79). In addition, people begin to artificially plant P. cyrtonema to increase income. Therefore, this study aims to investigate complete nutrient composition of the rhizomes and evalute the nutritional quality of different germplasm resources after artificial cultivation in Yuhang District, Hangzhou City, Zhejiang Province.

Re: 3) In my opinion the manuscript should include also the analysis of main active compounds of the plant.

Au: Thank you for your comment. Polysaccharides are the only quality indicators specified in the Chinese pharmacopoeia, which is one of the major components of carbohydrates. In addition, the present study focused on investigating the nutrient composition of Polygonatum cyrtonema Hua. Therefore, active coumpound polysaccharides were analyzed among the five germplasm.

Re: 4) The introduction should be rewritten. Add more information about the plant as a food.

Au: Thanks for your kind suggestions. We rewrote the introduction to provide more information about that Polygonatum was used as a food. With the increasing utilization of Polygonatum in food processing, it is necessary to further explore the nutritional composition of Polygonatum as well as to evaluate the nutritional quality of different germplasm resources after artificial cultivation. Please see details in the introduction section of the revised manuscript (Lines 32-87).

Re: 5) I couldn't find the novelity of the manuscript.

Au: Thanks for your suggestion. Polygonatum cyrtonema Hua is mainly consumed as food,but lack of comprehensive nutritional analysis components, which is important to develop new products and further guide consumer consumption. In this study, the total nutrient composition of P. cyrtonema was investigated, and the nutritional quality of the five germplasms after artificial cultivation was evaluated. We demonstrated that P. cyrtonema was a nutritious food, with a low starch content and greater amounts of carbohydrate, protein, Ca, Fe, and vitamin C in comparison with ginger, yam, and potatoes. After analyzed the composition of dietary fiber, P. cyrtonema was found to be high in dietary fiber with pectin and resistant starch being the major soluble components and hemicellulose being the dominant insoluble dietary fiber. The complex correlation relationships between 35 nutritional parameters, especially carbohydrate components and dietary fiber fractions in P. cyrtonema, were elucidated. Furthermore, the nutritional characteristics of the five germplasms and ten characteristic nutritional indicators were obtained by principal component analysis, which would provide valuable references for future development and utilization of Polygonatum.

In addition, we corrected the names of two samples from Hunan, changing from "Shaoyang" to "Yueyang". Thank you very much for your valuable comments for further improvement of our manuscript. And we also modified language and highlighted it with red color. Thank you for your positive evaluation and support for the MS.

Round 2

Reviewer 1 Report

Comments and Suggestions for Authors

The title of the work is ambiguous. To title should be re-written to improe clarity and make it more representative of the research work that was actually performed. For example, 

"Comprehensive evaluation of the nutritional properties of different germplasms of Polygonatum cyrtonema Hua"

Author Response

To Reviewer #1

Re: The title of the work is ambiguous. To title should be re-written to improe clarity and make it more representative of the research work that was actually performed. For example,

"Comprehensive evaluation of the nutritional properties of different germplasms of Polygonatum cyrtonema Hua"

Au: Thank you for your kind suggestion. The title was changed to "Comprehensive evaluation of the nutritional properties of different germplasms of Polygonatum cyrtonema Hua".

Thank you for reviewing our manuscript and offering valuable advice. All changes were highlighted in blue color.